# Profiling of Tryptophan Metabolic Pathways in the Rat Fetoplacental Unit during Gestation

**DOI:** 10.3390/ijms21207578

**Published:** 2020-10-14

**Authors:** Cilia Abad, Rona Karahoda, Petr Kastner, Ramon Portillo, Hana Horackova, Radim Kucera, Petr Nachtigal, Frantisek Staud

**Affiliations:** 1Department of Pharmacology and Toxicology, Faculty of Pharmacy in Hradec Kralove, Charles University, Akademika Heyrovskeho 1203, 500 05 Hradec Kralove, Czech Republic; abadmarc@faf.cuni.cz (C.A.); karahodr@faf.cuni.cz (R.K.); portillr@faf.cuni.cz (R.P.); horackha@faf.cuni.cz (H.H.); 2Department of Pharmaceutical Chemistry and Pharmaceutical Analysis, Faculty of Pharmacy in Hradec Kralove, Charles University, Akademika Heyrovskeho 1203, 500 05 Hradec Kralove, Czech Republic; kastner@faf.cuni.cz (P.K.); kucerar@faf.cuni.cz (R.K.); 3Department of Biological and Medical Sciences, Faculty of Pharmacy in Hradec Kralove, Charles University, Akademika Heyrovskeho 1203, 500 05 Hradec Kralove, Czech Republic; nachtigal@faf.cuni.cz

**Keywords:** fetal programming, tryptophan metabolism, rat model, fetal organs, pregnancy, placenta–brain axis

## Abstract

Placental homeostasis of tryptophan is essential for fetal development and programming. The two main metabolic pathways (serotonin and kynurenine) produce bioactive metabolites with immunosuppressive, neurotoxic, or neuroprotective properties and their concentrations in the fetoplacental unit must be tightly regulated throughout gestation. Here, we investigated the expression/function of key enzymes/transporters involved in tryptophan pathways during mid-to-late gestation in rat placenta and fetal organs. Quantitative PCR and heatmap analysis revealed the differential expression of several genes involved in serotonin and kynurenine pathways. To identify the flux of substrates through these pathways, Droplet Digital PCR, western blot, and functional analyses were carried out for the rate-limiting enzymes and transporters. Our findings show that placental tryptophan metabolism to serotonin is crucial in mid-gestation, with a subsequent switch to fetal serotonin synthesis. Concurrently, at term, the close interplay between transporters and metabolizing enzymes of both placenta and fetal organs orchestrates serotonin homeostasis and prevents hyper/hypo-serotonemia. On the other hand, the placental production of kynurenine increases during pregnancy, with a low contribution of fetal organs throughout gestation. Any external insult to this tightly regulated harmony of transporters and enzymes within the fetoplacental unit may affect optimal in utero conditions and have a negative impact on fetal programming.

## 1. Introduction

Throughout gestation, the catabolism of tryptophan (TRP), which is an essential amino acid, is critical for proper placental and fetal development [1] Therefore, the placental homeostasis of TRP, including transport and metabolism, must be tightly regulated [2]. In several organs, including the placenta, TRP is principally metabolized via the serotonin (5-HT) and kynurenine (KYN) pathways, and the relative flux of the resulting metabolites changes, depending on the stage of pregnancy and fetal demands [3]. TRP metabolism in the placenta has recently attracted considerable attention since various metabolites possess immunoprotective or neuroactive properties, with the ultimate effect on fetal neurodevelopment [4,5]. Subsequently, insults or challenges to placental TRP catabolism, such as pathologies (inflammation and diabetes), epigenetics, or polymorphisms of essential genes, have been associated with poor pregnancy outcomes and the development of mental or metabolic diseases later in life [1]. In addition, the timing of these insults is critical for fetal development [6] and, therefore, understanding TRP catabolic pathways in the placenta and fetal organs throughout gestation is of importance for identifying the biological roots of developmental origins of health and disease.

TRP catabolism along the 5-HT pathway gives rise to active metabolites, such as 5-HT and melatonin, which are important for fetal development and programming [7]. The key enzymes of this pathway are the 5-HT-producing tryptophan hydroxylase (TPH) and 5-HT-degrading monoamine oxidase (MAO) [8,9,10]. At the early stages of pregnancy, the placental synthesis of 5-HT is particularly important for successful blastocyst implantation, placentation, and decidualization [11,12]. In addition, 5-HT is crucial for the development of 5-HT-dependent organs, including the brain [13] and cardiac system [14]. However, the placental homeostasis of 5-HT remains controversial; current evidence suggests that in the early stages of pregnancy, maternal 5-HT is critical for embryonic development [15]; nevertheless, at E10.5 of mouse [16] and 11 weeks of human gestation [17] this role is taken up by the placenta’s own 5-HT neosynthetic capacity. With advancing gestation, however, the fetus is able to synthesize its own 5-HT from maternally-derived TRP [18]. Moreover, we have recently demonstrated that rat and human term placentas rapidly extract 5-HT from the fetal circulation into trophoblast cells by the organic cation transporter (OCT3/SLC22A3), where it is degraded by MAO-A [19]. Since both hyper- and hypo-serotonemia are detrimental for fetal development [20] and an excess of 5-HT is lethal due to its vasoconstrictive properties [21] placenta and fetal organs must tightly regulate the expression and activity of the key enzymes and transporters involved in 5-HT homeostasis in the fetoplacental unit during the whole period of gestation.

Concurrently, the second branch of placental TRP metabolism—the KYN pathway—plays an essential role in allogeneic fetal rejection and is important for achieving immunotolerance for the fetus [22,23]. The human placenta functionally expresses indoleamine 2,3-dioxygenase (IDO)—the initial and rate-limiting enzyme for the KYN pathway [24]. Moreover, the gene expression of all enzymes involved in the KYN pathway has been reported [25]. Neuroactive in nature, several end-products of KYN metabolism may be released into the fetal circulation [26] and affect fetal development. Specifically, through their opposing activity on the N-methyl-D-aspartate (NMDA) receptor, KYN metabolites affect neuronal functions. While kynurenic acid (KYNA) is an antagonist with neuroprotective properties, quinolinic acid (QUIN) acts as an agonist with neurotoxic activity [27]. Additionally, KYN metabolites such as 3-hydroxykynurenine, anthranilic acid, and 3-hydroxyanthranilic acid have been reported to exert cytotoxic activity [28,29,30,31]. Collectively, malfunctions of the KYN pathways in rats have been linked to poor pregnancy outcomes [1,32] and it has been suggested that placenta-derived KYN metabolites may be involved in the etiology of prenatal brain damage [32].

We have recently reported that the human placental catabolism of TRP is subject to developmental changes during pregnancy and trophoblast differentiation [33]. In this study, we hypothesized, that apart from the placenta, fetal organs also contribute to overall TRP homeostasis in the fetoplacental unit. Since experiments in pregnant women are limited due to ethical and technical reasons, current research depends on the use of alternative approaches, including experimental animals. Therefore, here, we used the Wistar rat as the most suitable model for placental TRP metabolism in health and disease [3] in order to provide in-depth evidence on the regulation of TRP homeostasis in the fetoplacental unit during gestation.

## 2. Results

### 2.1. Basal Levels of TRP, KYN, KYNA, 5-OH-TRP, and 5-HT in the Maternal Plasma and Placenta

We first analyzed the metabolite concentrations in the rat placenta and maternal plasma on different gestation days (GDs). Maternal plasma levels of TRP, KYN, KYNA, 5-OH-TRP, and 5-HT remained stable during gestation (Table 1). Likewise, metabolite/precursor ratios, specifically KYN/TRP, KYNA/KYN, and 5-OH-TRP/TRP ratios, remained steady, suggesting that the metabolic activity displayed by maternal organs is not significantly affected by advancing pregnancy; nevertheless, considerable interindividual variability was detected.

In placental tissues, we observed a significant increase in TRP concentrations (Figure 1A), while 5-OH-TRP concentrations (Figure 1B) and the 5-OH-TRP/TRP ratio (Figure 1D) decreased significantly during gestation, suggesting a decrease in TRP catabolism along the 5-HT pathway towards term. Interestingly, placental concentrations of 5-HT increased, reaching top levels at GD 18 (Figure 1C). KYNA levels in the placenta increased during gestation (Figure 1F), whereas KYN concentrations (Figure 1E) and the KYN/TRP ratio (Figure 1G) decreased from GD 12 to GD 21.

### 2.2. Gene Expression of Enzymes and Transporters Involved in TRP Metabolic Pathways in the Rat Placenta

Next, in high-throughput analysis, we examined the gene expression of 22 target genes in five rat concepti samples (GD 12) and 16 rat placenta samples at different stages of gestation (*n* = 5 for GD 15 and 18; *n* = 6 for GD 21). Figure 2A displays a heatmap analysis with an overview of the expression profiles, as well as hierarchical clustering applied to group samples with similar expression levels. Based on the Mann–Whitney test, comparing only the placenta tissue, several enzymes/transporters exhibited differential expression levels during gestation (Figure 2B–D). Statistical analysis against the concepti samples was not performed, as it represents a mixed tissue (embryonic and extra-embryonic).

*Mao-a*, *Slc3a2*, and *Slc6a4* showed statistically significant upregulation at GD 18 and 21, compared to GD 15 (Figure 2B,C). On the other hand, *Qprt*, *Asmt*, *Slc7a8*, and *Aanat* revealed upregulation at GD 21, compared to both GD 15 and 18 (Figure 2C,D). The increased gene expression of *Kynu*, *Haao*, *Slc18a2*, and *Slc7a5* is evident at all stages of pregnancy tested (Figure 2B–D).

### 2.3. Absolute Quantification of Tph1, Mao-a, Slc6a4, Slc22a3, and Ido2 Gene Expression in the Rat Placenta during Gestation

The absolute quantification of transcripts/ng RNA was performed using ddPCR analysis on five placental samples from each gestational age. The rate-limiting enzyme of the 5-HT pathway—*Tph1*—was found to be expressed in very low amounts at all stages of gestation (<1 transcript/ng RNA) (Figure 3A). In contrast, *Mao-a* expression peaked on GD 18 and decreased at GD 21 (Figure 3D). Interestingly, we found that *Ido1* is not expressed in the rat placenta. Therefore, we evaluated *Ido2* expression as the main isoform of *Ido* in rats. *Ido2* levels were found to be ~9 transcripts/ng RNA at GD 15 and significantly decreasing at GD 21 (Figure 3G). As for 5-HT transporting proteins, *Slc6a4* revealed significant upregulation during gestation (Figure 3J), while *Slc22a3* remained unchanged, although at several-fold higher levels when compared to those of *Slc6a4* (Figure 3L).

It is worth noting that at GD 15, the ratio of *Mao-a/Tph1* transcripts is approximately 1198, whereas at GD 21, it increases to 5377, suggesting an increasing importance of placental 5-HT degradation vs. 5-HT synthesis towards the end of gestation.

### 2.4. Protein Analysis of TPH, MAO, SLC6A4, SLC22A3, and IDO in the Rat Placenta during Gestation

To investigate changes in protein expression, quantitative western blot analyses using specific antibodies for TPH, MAO, SLC6A4, SLC22A3, and IDO were performed in rat placenta homogenate samples from GD 15, 18, and 21 (*n* = 5). We observed a significant decrease in TPH protein expression in the terminal phase of rat pregnancy (GD 21) compared to GD 15 (Figure 3B). On the other hand, IDO protein expression showed an increased trend toward GD 21, compared to GD 15 and 18 (Figure 3H). Altogether, the results of western blot analysis, in agreement with gene expression data, suggest preferential TRP metabolism towards the KYN pathway at the final stages of rat pregnancy.

MAO protein expression also corresponded with mRNA levels and reached a peak at GD 18 (Figure 3E). SLC6A4 protein expression increased significantly from mid-pregnancy to GD 21 (Figure 3K), whereas SLC22A3 protein expression dropped in the terminal stages of rat pregnancy (Figure 3M).

### 2.5. In Vitro Functional Assays of TPH, MAO, and IDO in the Rat Placenta

The enzymatic activity of TPH, which is a 5-HT-producing enzyme, was relatively stable during gestation, showing about a 50% and 30% increase at GD 18 and 21, respectively, when compared to GD 15 (Figure 3C). In contrast, the enzymatic activity of 5-HT-degrading MAO increased by almost 400% towards GD 18 and 21 (Figure 3F). This metabolism was completely inhibited by 100 µM phenelzine—a MAO inhibitor—indicating that 5-HT degradation was exclusively due to the activity of this oxidase [19]. Lastly, IDO activity approximated 19.7 nmol KYN/µg protein per min and no statistically significant differences were observed between GD 15, 18, and 21 (Figure 3I).

### 2.6. Immunohistochemical Localization of TPH, MAO, IDO, and SLC6A4 in the Rat Placenta at Different Stages of Gestation

Immunohistochemical analyses were performed to localize TPH, MAO, IDO, and SLC6A4 in the rat placenta during middle- (GD 15) and late-stage (GD 18 and 21) pregnancy. At GD 15, TPH expression was mostly detected in cytotrophoblast cells and fetal capillaries inside the labyrinth zone (Figure 4A); additionally, positivity in blood leukocytes was observed. At GD 18, no positivity for TPH in trophoblastic giant cells and the trophospongium area was detected. In contrast, more intense immunostaining was visible in the cytotrophoblast cells in the labyrinth zone (Figure 4B). Similarly, cytotrophoblast cells and fetal capillaries inside the labyrinth zone at GD 21 were well-stained for TPH (Figure 4C).

MAO expression was only detected in cytotrophoblast/syncytiotrophoblast cells in the area of the forming labyrinth at GD 15 (Figure 4D). No reactivity was detected in the trophoblast giant cell area. A similar staining pattern was visible at GD 18, where MAO was detected in the syncytiotrophoblast within the labyrinth area (Figure 4E). Moreover, at GD 21, MAO was predominantly visible in the inner layers of the syncytiotrophoblast (layer II and III) (Figure 4F). No positivity for MAO was visible in the trophospongium area at GD 18 and GD 21.

At GD 15, the IDO immunostaining pattern was weakly detected in trophoblastic giant cells, with no reaction in the trophospongium area (Figure 4G). The strongest positivity was visible in cytotrophoblast cells in the forming labyrinth zone (Figure 4G). Furthermore, IDO positivity was detected in cytotrophoblast cells and fetal endothelial cells within the labyrinth zone (Figure 4H) at GD 18. Similar immunostaining for IDO in the labyrinth zone, specifically in cytotrophoblast cells and fetal endothelial cells, was visible at GD 21 (Figure 4I).

Lastly, at GD 15, SLC6A4 expression was detected in the trophoblast giant cell area; nevertheless, no reactivity was visible in the trophospongium area (Figure 4J). Weak SLC6A4 expression was detected in cytotrophoblast/syncytiotrophoblast cells in the forming labyrinth (Figure 4J). At GD 18, SLC6A4 expression was mainly detected in syncytiotrophoblast cells in the labyrinth area (no expression in the trophospongium area) (Figure 4K). The SLC6A4 expression staining pattern was similar to that observed for MAO at GD 21, with positivity detected in the inner layers of the syncytiotrophoblast (layer II and III) (Figure 4L).

### 2.7. Gene Expression of Key Enzymes/Transporters of TRP Metabolic Pathways in Fetal Organs

The rat fetal brain, liver, heart, lungs, and intestine were examined for the gene expression of key enzymes/transporters of 5-HT and KYN pathways. Specifically, *Ido1*, *Ido2*, *Tdo2*, *Tph1*, *Tph2*, *Mao-a*, *Mao-b*, *Slc6a4*, *Slc7a5*, and *Slc22a3* were analyzed. *Ido1* was not detected in any of the fetal tissues tested, whereas *Tph2* was not detected in the fetal liver. All other genes were expressed, showing several differential expression patterns at GD 18 and 21 (Figure 5). In particular, the following genes were upregulated during gestation in respective tissues: *Slc22a3* in the fetal brain (Figure 5A); *Slc6a4* and *Mao-b* in the fetal intestine (Figure 5B); and *Tdo2* and *Ido2* in the fetal liver (Figure 5C). The following genes were downregulated during gestation in respective tissues: *Tph2* and *Ido2* in the fetal heart (Figure 5D) and *Slc22a3* in fetal lungs (Figure 5E).

### 2.8. Absolute Quantification of Tph, Mao, and Ido in Fetal Organs at Term

At GD 21, the expression of the rate-limiting enzyme of the 5-HT pathway—*Tph1*—showed significantly higher levels in the fetal intestine compared to the fetal brain (*p* = 0.0043), lungs (*p* = 0.0043), and liver (*p* = 0.0043) (Figure 6A). Interestingly, at the same gestational age, *Tph1* expression in the fetal brain (*p* = 0.0079), intestine (*p* = 0.0043), and lungs (*p* = 0.0079) was significantly higher compared to the levels observed in placental tissue (Figure 6A), suggesting that, at later stages of rat pregnancy, the fetus takes over 5-HT synthesis for its own needs.

Similarly, *Mao-a* was expressed at high amounts in all fetal organs; nevertheless, the fetal intestines showed a significantly higher expression compared to fetal lungs (*p* = 0.0087) and the liver (*p* = 0.026) (Figure 6A). In addition, at GD 21, the rat placenta (Figure 6A) showed similar *Mao-a* expression levels to the fetal brain (*p* = 0.69), intestine (*p* = 0.43), and liver (*p* = 0.052), whereas the expression in fetal lungs was significantly lower (*p* = 0.016).

Regarding the KYN pathway, *Ido2* expression was highly evident in the fetal brain, intestine, and liver (Figure 6A), with the levels in the fetal liver being 10 times higher than those in the fetal brain and intestine and 100 times higher than those in fetal lungs. In general, the rat placenta at GD 21 (Figure 6A) reveals similar levels of the expression of *Ido2* compared to the fetal brain (*p* = 0.22), fetal intestine (*p* = 0.33), and fetal liver (*p* = 0.15) and a significantly higher expression compared to fetal lungs (*p* = 0.0075).

### 2.9. Functional Activity of TPH, MAO, and IDO in Fetal Organs at Term

The fetal brain and intestine showed the highest TPH enzymatic activity compared to fetal lungs and the liver (Figure 6B). Nonetheless, at GD 21, these levels were comparable to those of the placenta. On the other hand, the fetal brain showed the highest capacity to metabolize 5-HT by MAO (approximately 25% of added 5-HT) when compared to the fetal intestine (13% of added 5-HT), lungs (10% of added 5-HT), and liver (8.5% of added 5-HT) (Figure 6B). The levels of MAO activity in the fetal brain were comparable to those observed in the rat placenta at the same stage of pregnancy (33% of added 5-HT). Interestingly, the MAO activity in the rat placenta at GD 21 was significantly higher compared to the fetal intestine (*p* = 0.016), lungs (*p* = 0.016), and liver (*p* = 0.0079).

At GD 21, the rat placenta displayed the highest IDO enzymatic activity, compared to any of the fetal organs, specifically the brain (*p* = 0.0079), intestine (*p* = 0.0079), lungs (*p* = 0.036), and liver (*p* = 0.0079) (Figure 6B).

## 3. Discussion and Conclusions

During pregnancy, the TRP demand at the materno–fetal interface increases as a result of fetal growth and development [3]. Since TRP is an essential amino acid, the placenta and fetus are dependent on its maternal intake and placental transport from the maternal circulation. In this paper, we provide in-depth evidence on the TRP metabolic pathways in the fetoplacental unit during gestation. We demonstrate that the dynamics of both 5-HT and KYN pathways are largely driven by the demands of the developing fetus that, towards the end of gestation, is fully capable of metabolizing TRP to 5-HT and partially to KYN. Importantly, we show that the placenta is a fundamental component in maintaining proper metabolite levels (specifically 5-HT), with an efficiency analogous to that of the fetal brain.

Consistent with previous reports in rats [34,35], we detected stable TRP concentrations in maternal plasma during gestation; in addition, we have shown that the placental TRP content increases with advancing gestation. This increase correlates nicely with the upregulated expression of transport proteins involved in the placental uptake of TRP, namely L-amino acid transporter-1 (*Slc7a5*, LAT1) and 2 (*Slc7a8*, LAT2), and the 4F2 heavy chain (*Slc3a2*) crucial for the functional activity of LAT1/2. We propose that these changes reflect physiological adaptations to increased placental and fetal demands for TRP as a protein component, but also precursor, for neurotransmitters, hormones, and other bioactive molecules (Figure 7).

Similar to TRP, we observed an increase in placental levels of 5-HT, despite its steady concentrations in maternal blood. This phenomenon was first described by Robson and Senior, who speculated that fetal and/or placental 5-HT synthesis may account for the changes [36]. To investigate this ‘paradox’ in more detail, we analyzed the expression and activity of key components responsible for the synthesis (TPH), degradation (MAO-A), and transport (SERT and OCT3) of 5-HT in both placenta and fetal organs. Since 5-OH-TRP is an intermediate in 5-HT production from TRP, the decrease in the 5-OH-TRP and 5-OH-TRP/TRP ratio observed in our study indicates decreased placental 5-HT synthesis towards term. This finding is further supported by the lower levels of *Tph1* transcripts and TPH protein at the final stages of rat pregnancy. Next, we hypothesized that the increase in placental 5-HT must result from fetal sources, as current literature shows that, at later stages of pregnancy, the fetus is able to synthesize its own 5-HT from maternally-derived TRP [37]. Indeed, all fetal organs evaluated in our study (i.e., the intestine, brain, lungs, and liver) showed absolute levels of *Tph1* transcripts higher than those of the term placenta. Additionally, we detected TPH enzymatic activity in the fetal brain and intestine, confirming and identifying the functional 5-HT synthetic machinery in the fetus at term.

Regarding the transport of 5-HT into the placenta, we have recently proposed that rat and human term placentas effectively extract 5-HT from both maternal and fetal circulations for subsequent metabolism by MAO-A [19]; this clearance mechanism is important for protecting placental circulation against hyperserotonemia and subsequent vasoconstriction. The expression of SERT in terms of both mRNA and protein levels increased significantly towards GD 21, while that of OCT3 remained relatively stable. However, when comparing the absolute transcript levels of these two transporters, we saw a considerably higher expression of OCT3 when compared with SERT at all GDs tested. This indicates the ‘higher activity’ of OCT3 in the fetoplacental unit, as this polyspecific, low-affinity but high-capacity transporter [38] is responsible for the transport of not only 5-HT, but also other monoamines (our unpublished data) and toxins from the fetus [39,40] Therefore, we can conclude that increased levels of placental 5-HT at term are mainly due to increased transporter-mediated uptake of 5-HT from both maternal and fetal circulations. Correspondingly, the increased MAO expression and activity observed in our study and human [33], murine [41], and rat [42,43] placentas suggest that extracted 5-HT is efficiently degraded to inactive 5-hydroxyindoleacetic acid. Supportive of this placental clearance hypothesis, we observed the co-localization of SERT and MAO in syncytiotrophoblast cells, specifically, layer II and III within the labyrinth area. A previous study by our team reported similar immunostaining patterns for OCT3 in the labyrinth zone [44] and the co-localization of OCT3 and MAO has also been shown in the mouse placenta [45]. Taken together, SERT, OCT3, and MAO-A seem to be the key components of placental handling of the 5-HT pathway of TRP metabolism and play a critical role in maintaining 5-HT homeostasis in the fetoplacental unit (Figure 7).

Recent literature stresses the importance of cross-talk between the placenta and fetal organs, pointing to, e.g., the placenta–brain axis [46,47] or placenta–heart axis [48,49]. This mutual communication and collaboration between the placenta and fetal organs is of paramount importance for proper in utero development and fetal programming. Therefore, to investigate whether other mechanisms, driven by the fetus, can control the fetal circulating levels of 5-HT, we also analyzed the expression and activity of TPH and MAO in the fetal organs and compared them with those of the placenta. The results show that, during the prenatal period, the placenta and fetal brain are the most metabolically active organs in 5-HT degradation by MAO, followed by the intestine, lungs, and liver. These findings correspond well with reports by other groups for rats and humans, where placental MAO-A activity exceeds that of the lungs and liver [50,51]. Consequently, the interplay between placental and fetal MAO activity seems fundamental for regulating 5-HT circulating levels in the fetoplacental unit and wiring the placenta–brain axis for proper neurodevelopment of the fetus.

In mammals, the KYN pathway represents the major catabolic route of TRP metabolism in many tissues, including the placenta [3]. The first and rate-limiting step of TRP metabolism along the KYN pathway is catalyzed by IDO1, which is a ubiquitous enzyme with the highest expression and activity in the placenta, lungs, and intestine. It was first described in the human placenta by Yamazaki et al. in 1985 [24], almost three decades later, two other enzymes were attributed to TRP degradation in the placenta—IDO2 and tryptophan 2,3-dioxygenase (TDO) [1]. Their relative contribution to TRP metabolism and dynamics along the course of gestation is still not fully elucidated. In our study, we found that rat placenta and fetal organs do not express the *Ido1* gene; instead, *Ido2* is predominant. Nevertheless, its protein localization in the vascular endothelium coincides with that of IDO1 in the human placenta [52]. Additionally, we localized IDO expression in cytotrophoblast cells which corresponds with our recent findings in primary trophoblast cells isolated from human term placenta [33]. We observed an increase in IDO protein expression and stable activity during gestation, which is in good agreement with previous reports [5,52]. However, despite steady IDO activity, the placental content of KYN decreased significantly towards term. This can be explained by KYN transport to fetal circulation since fetal organs can utilize KYN for the synthesis of KYNA, which as an important neuroprotective metabolite [32]. Alternatively, this decrease might also be attributed to the rapid utilization of KYN for the synthesis of other metabolites inside the placenta; although statistically insignificant, we detected a steady increase in the KYNA placental content from GD 15 to 21. Nevertheless, placental handling of KYN and KYNA is not unified in the current literature [32,53,54] and requires further investigation.

During the prenatal period in the rat, IDO activity was most pronounced in the placenta, followed by fetal lungs and intestine. On the other hand, the fetal liver showed the highest *Ido2* transcript numbers and significant upregulation of *Ido2* and *Tdo2* gene expression from GD 18 to 21. Nevertheless, contrary to the young rat liver, which shows the highest IDO activity and KYN content [55], KYN production in the fetal liver via IDO at term is notably lower. Previous studies have also shown that TDO activity is absent in the liver of fetuses and young rats [56,57]. Collectively, these findings suggest that although the enzymes are highly expressed, the fetal liver is not yet fully functional for KYN production. Indeed, the maturation of hepatocytes in the prenatal period is gradual, beginning at E13 in mice, and continuing until after birth [58].

TRP catabolism within the fetoplacental unit is a very complex network of numerous genes and transcription factors controlling the production of several active metabolites of both 5-HT and KYN pathways. Therefore, any insult, including pathological conditions, environmental factors, medication, and/or epigenetics during pregnancy, may significantly affect its orchestration, resulting in suboptimal in utero conditions. Specifically, polymorphisms of its decisive components, including SERT [59,60], OCT3 [61], TPH [62], IDO [63] and MAO [64], have all been linked to poor mental health outcomes. Additionally, environmental factors (including prenatal stress, diet, and smoking) modify MAO-A expression and function [64]; the KYN pathway is particularly susceptible to inflammation [65] and hypoxia [66,67] during pregnancy. Considering that the timing of the insults is an important variable in the fetal programming and offspring outcome [68,69] knowledge of these physiologically occurring changes is critical for identifying mechanisms and pathways involved at precise time-windows during gestation.

In conclusion, our results obtained from the Wistar rat demonstrate that under physiological conditions, placental TRP metabolism to 5-HT is crucial up to mid-gestation. Afterwards, the fetus is capable of 5-HT synthesis and becomes independent of maternal and/or placental sources. Moreover, at term, the placenta and fetal organs orchestrate 5-HT homeostasis via the activity of transporters and metabolizing enzymes, likely to prevent hyper/hypo-serotonemia in the fetoplacental unit. On the other hand, the placental production of kynurenine increases during pregnancy, with a low contribution of fetal organs throughout gestation (Figure 7). Importantly, our data obtained from the rat placenta are in close agreement with those observed in humans [33], confirming Wistar rat as an appropriate model for further studies on TRP homeostasis in the fetoplacental unit.

## 4. Materials and Methods

### 4.1. Chemicals and Reagents

Serotonin hydrochloride, L-tryptophan, and phenelzine (MAO inhibitor) were purchased from Sigma-Aldrich (St. Louis, MO, USA). Bicinchoninic acid assay (BCA assay) reagents were purchased from Thermo Scientific (Rockford, IL, USA). Tri Reagent solution was obtained from the Molecular Research Centre (Cincinnati, OH, USA). All other chemicals were of analytical grade.

### 4.2. Rat Placenta and Fetal Organ Sample Collection

The experiments were performed using female Wistar rats (approximately 5–6 months old) purchased from Biotest s.r.o. (Czech Republic). All experiments were approved by the Ethical Committee of the Faculty of Pharmacy in Hradec Kralove (no. MSMT-4312/2015-8; Charles University, Czech Republic) and applied procedures were compliant with the ARRIVE guidelines and have been carried out in accordance with the U.K. Animals (Scientific Procedures) Act, 1986 and associated guidelines of EU Directive 2010/63/EU for animal experiments. Descriptive statistics of the rats used in the study can be found in Table 1. The animals were maintained in cages, with constant room temperature, low noise, and 12L:12D standard conditions. All rats received food and water ad libitum. Gestation day (GD) 1 was established upon the detection of a copulatory plug of sperm after overnight mating. Experiments were performed at GD 12, 15, 18, and 21. At GD 12, the rat placenta is technically indistinguishable from the embryo. Therefore, we present these results separately as concepti. Similarly, as the fetal organ collection at earlier gestational ages provides insufficient material for accurate analysis, the experiments on fetal organs were limited to GD 18 and 21.

Rats were anesthetized with pentobarbital at a dose of 40 mg/kg administered into the tail vein. Placental tissue and fetal organs (kidney, liver, brain, intestine, heart, and lung) were dissected and used for homogenate preparation and expression analysis. Samples were stored at −80 °C until analysis.

### 4.3. RNA Isolation, Reverse Transcription, and Quantitative PCR Analysis

Total RNA was isolated from weighed tissue samples using Tri Reagent solution, according to the manufacturer’s instructions. The purity of the isolated RNA was checked by the A260/A280 ratio, whereas the A260/230 ratio was used to evaluate contamination by organic solvents. Absorbance ratios were measured using a NanoDrop™ 1000 Spectrophotometer (Thermo Fisher Scientific, Waltham, MA, USA) and the A260 measurement was used for calculation of the total RNA concentration. RNA integrity was confirmed by electrophoresis on a 1.5% agarose gel. Reverse transcription (RT) was performed using the iScript Advanced cDNA Synthesis Kit and T100^TM^ Thermal Cycler (Bio-Rad, Hercules, CA, USA).

Quantitative PCR (qPCR) analysis of gene expression in rat placenta and fetal organs was performed using QuantStudio^TM^ 6 (Thermo Fisher Scientific, Waltham, MA, USA). cDNA (12.5 ng/µl) was amplified in a total reaction volume of 5 µL/well using the TaqMan^®^ Universal Master Mix II without UNG (Thermo Fisher Scientific, Waltham, MA, USA) and predesigned TaqMan^®^ Real Time Expression PCR assays (listed in Appendix A). Each sample was amplified in triplicate, using the following PCR cycling profile: 95 °C for 10 min, followed by 40 cycles at 95 °C for 15 s and 60 °C for 60 s.

Stable expression of the reference gene was evaluated, and gene expressions were normalized against two predesigned reference genes (listed in Appendix A) using the ΔCt method, whereby ΔCt = Ct_ref_ − Ct_target_. These values were used to generate a gene expression heat map, through the freely available web server Heatmapper (http://www.heatmapper.ca/) [70]. Hierarchical clustering (Average linkage, Euclidean distance) was applied to group samples with similar expression levels. The scatter plot was constructed in GraphPad Prism 8.3.1 software (GraphPad Software, Inc., San Diego, CA, USA), using the average 2^ΔCt^ values for different gestational days.

### 4.4. Droplet Digital PCR Assay

Duplex ddPCR analysis of *Slc6a4* and *Slc22a3* in the rat placenta and *Mao-a*, *Tph1*, and *Ido2* in rat placenta and fetal organs was performed as described previously [33]. Using the duplex feature, we were able to absolutely quantify the gene expression of target and reference genes simultaneously. Briefly, the duplex reaction mixture consisted of 10 µL of ddPCR™ Supermix for Probes (Bio-Rad, Hercules, CA, USA), 1 µL of each of the predesigned probe assays (FAM and HEX) (listed in Appendix A), and 0.5 µL of cDNA (50 ng/µL), in a total volume of 20 µL. Droplets were generated using a QX200 Droplet Generator and subsequently amplified to end-point using a T100™ Thermal Cycler in the following conditions: Single cycle of 95 °C for 10 min, followed by 40 cycles of 94 °C for 30 s and 60 °C for 1 min and a single cycle of 98 °C for 10 min. Droplet counting was performed in a QX200™ Droplet Reader and the concentration of the target gene was calculated using the QuantaSoft™ Software. For final data evaluation, wells with droplet numbers of less than 13,000 were excluded. Results are reported as the number of transcripts/ng of transcribed RNA. The QX200™ Droplet Digital™ PCR System, T100™ Thermal Cycler, and all consumables and reagents were obtained from BioRad, Hercules, CA, USA (unless otherwise stated).

### 4.5. Preparation of Tissue Homogenates

The placentas (with amniochorion and chorionic plate removed) and fetal organs were washed with 0.9% NaCl, weighed, cut into small pieces, and homogenized in a solution (3 mL/g) containing 250 mM sucrose, 50 mM Tris-HEPES (pH = 7.2), 5 mM EGTA, 5 mM EDTA, and 1 mM PMSF. The homogenates were filtered through gauze and centrifuged at 800× *g* for 10 min. All procedures were performed at 4 °C. Supernatants were collected and stored in the freezer at −80 °C until use. The protein concentration was determined using the Pierce^TM^ BCA protein assay kit.

### 4.6. Western Blot Analysis

Aliquots of placenta homogenate (80 µg total protein) were mixed with loading buffer under reducing conditions [71], heated at 96 °C for 5 min; and separated by SDS-PAGE on 10% (SERT, OCT3), 12.5% (MAO) or 15% (IDO and TPH) polyacrylamide gels. Electrophoresis was performed at 120 V and proteins were transferred to PVDF membranes (Bio-Rad, Hercules, CA, USA). The membranes were blocked in 20 mM Tris-HCl pH 7.6, 150 mM NaCl, and 0.1% Tween 20 (TBS-T) containing 5% BSA for 1 h at room temperature and washed with TBS-T buffer. Incubation with primary antibodies (listed in Appendix A) was performed overnight at 4 °C against SLC6A4, SLC22A3, MAO, IDO, and TPH. After washing with TBS-T buffer, the membranes were incubated with a specific secondary antibody (listed in Appendix A) for 1 h at room temperature. Membranes were developed using Amersham^TM^ ECL^TM^ Prime (GE Healthcare Life Science, Marlborough, MA, USA). The band intensity was visualized and quantified by densitometric analysis using the ChemiDoc^TM^ MP, Imaging system (Bio-Rad, Hercules, CA, USA). To ensure the equal loading of proteins, membranes were probed for β-actin and specific secondary antibodies (listed in Appendix A).

### 4.7. 5-HT Metabolism by MAO in the Placenta and Fetal Organs

For MAO activity assays, 180 µL placenta or fetal organs homogenate (1.5–2 mg/mL) was preincubated for 5 min at 37 °C, with or without phenelzine (100 µM). The reactions were started by adding 20 µL of 5-HT (0.5 mM). After 60 min, the reaction was stopped by adding 40 µL of HClO_4_ (3.4 M) and placed on ice for 5 min [72]. Samples were centrifuged at 5000× *g* for 10 min, and the supernatant was used for 5-HT determination by HPLC. Results are expressed as metabolized 5-HT (% of initially added 5-HT).

### 4.8. TRP Metabolism by IDO in the Placenta and Fetal Organs

IDO activity in rat placenta homogenates or fetal organs was determined by measuring the quantity of KYN produced from TRP, according to the method of Takikawa et al. [73]. Briefly, 200 µL of placenta homogenate was incubated for 30 min at 37 °C with 800 µL of the incubation media (50 mM potassium phosphate buffer pH 6.5, 20 mM ascorbate, 0.01 mM methylene blue, 100 units/mL catalase, with and without 0.4 mM L-TRP). The reaction was stopped by adding 200 µL of 30% trichloroacetic acid, and further incubation at 50 °C for 30 min to hydrolyze N-formyl KYN to KYN. The reaction mixture was centrifuged at 3000× *g* for 20 min, at 20 °C, and 800 µL supernatant was collected for KYN measurement by HPLC. The IDO enzymatic activity was calculated as the difference between the amount of KYN produced in the media with/without L-TRP. The results are expressed as the nmol KYN/µg protein per min.

### 4.9. TRP Metabolism by TPH in the Placenta and Fetal Organs

TPH activity was determined according to the method described by Goeden et al. [74]. The incubation media (800 µL) contained (final concentrations) 50 mM Tris buffer pH 7.5, 1 mM EGTA, 100 units/mL catalase, 0.1 mM ammonium iron (II) sulphate, and 0.1 mM tetrahydrobiopterin (BH4, a cofactor required for TPH activity), with or without 0.25 mM L-TRP. The incubation medium was pre-incubated for 5 min at 37 °C and the reaction was initiated by adding placenta or fetal organ homogenates (200 µL) and incubated for 30 min at 37 °C. The reaction was terminated by adding 200 µL of HClO_4_ 0.2 mM with 100 µM EDTA. Samples were kept on ice for 15 min for complete protein denaturation and then centrifuged at 21,000× *g* for 15 min. Supernatants (800 µL) were collected for 5-hydroxy-l-tryptophan (5-OH-TRP) determination by HPLC, which is an intermediate in 5-HT production. The results were calculated as the difference between the amount of 5-OH-TRP liberated in tubes with/without L-TRP, and are expressed as nmol 5-OH-TRP/µg protein per min.

### 4.10. Extraction of TRP and Its Metabolites from Biological Samples

TRP, KYN, KYNA, 5-OH-TRP, and 5-HT were measured in maternal plasma and placental homogenate samples according to the method described by Goeden et al. [32]. Briefly, 150 µL of 25% perchloric acid was added to 600 µL of each sample and incubated on ice for 10 min. Precipitated protein was removed by centrifugation at 16,000× *g* for 15 min. The supernatant was collected and used for HPLC analysis.

### 4.11. HPLC Analysis of TRP, 5-HT, 5-OH-TRP, KYN, and KYNA

The HPLC analyses were performed using a Shimadzu LC20 Performance HPLC chromatograph (Shimadzu, Kyoto, Japan) equipped with a UV and fluorescence detector. For the chromatographic separation of all tested compounds simultaneously, Phenomenex Kinetex 5 µm C18 100 Å 150 × 3 mm with a guard column was used. Isocratic elution at a flow rate of 0.5 mL/min was performed with a mobile phase consisting of 0.1 M acetic acid, pH 4.5 (adjusted with NaOH), and methanol (97:3, *v*/*v*). All analytes were eluted within 9 min.

Excitation and emission wavelengths of fluorescence detector were set for individual compounds: 275/333 nm for 5-OH-TRP from 0 to 3.1 min and 280/334 nm for 5-HT and TRP from 3.1 min. The wavelength of the UV detector was set to 369 nm for KYN from 0 to 6.5 min and 244 nm for KYNA from 6.5 min.

### 4.12. Immunohistochemical Characterization of IDO, TPH, MAO, and SERT in the Rat Placenta

Preparation of rat placental tissue at GD 15, 18, and 21 was performed as described previously [44]. For all antibodies, the antigen was unmasked by heating the specimens in sodium citrate buffer (pH 6.0) in a microwave oven at 750W. Slides were incubated with primary antibody against MAO, SLC6A4, IDO, and TPH (listed in Appendix A). Subsequently, all slides were developed with the ImmPRESS^®^ HRP Goat Anti-Rabbit IgG Polymer Detection Kit, Peroxidase (Cat# MP-7451; Vector Laboratories, Burlingame, CA, USA), for 30 min. The reaction was visualized using diaminobenzidine (DAB substrate-chromogen solution; Dako, Carpinteria, CA, USA) and the sections were counterstained with hematoxylin. The specificity of the immunostaining was assessed by staining with nonimmune isotype-matched immunoglobulins. Slides were examined using computer image analysis (light microscope Nikon Eclipse E200, Japan; Digital camera Nikon DS-Fi3, Japan; NIS software, version 5.402, Laboratory Imaging, Prague, Czech Republic).

### 4.13. Statistical Analysis

Experimental outcomes were assessed using the non-parametric Mann–Whitney (when comparing two groups) or Kruskal–Wallis tests, followed by Dunn’s multiple comparisons test (when comparing more than two groups). All analyses were implemented in GraphPad Prism 8.3.1 software (GraphPad Software, Inc., San Diego, USA). Asterisks in the figures indicate significance levels: * (*p* ≤ 0.05), ** (*p* ≤ 0.01), and *** (*p* ≤ 0.001).

## Figures and Tables

**Figure 1 ijms-21-07578-f001:**
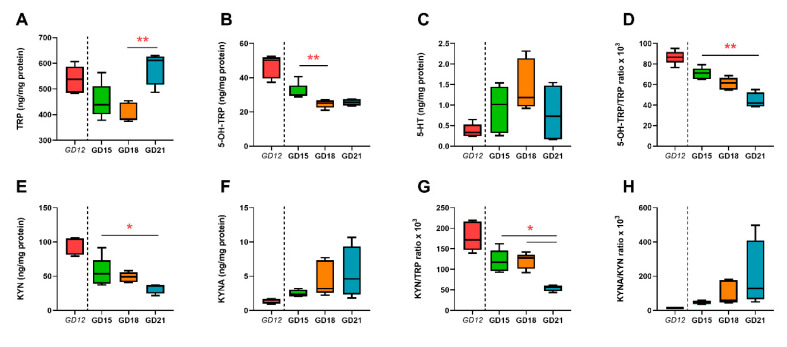
Basal tryptophan (TRP) metabolite levels in the rat placenta during gestation. Concentration of TRP **(A)** and its main metabolites—5-hydroxy-l-tryptophan (5-OH-TRP) (**B**), serotonin (5-HT) (**C**), kynurenine (KYN) (**E**), and kynurenic acid (KYNA) (**F**)—was evaluated on various gestation days (GD); ratios of metabolite to precursor are also shown (**D,G,H**). It is important to note that at GD 12, the rat placenta is technically indistinguishable from the embryo. Therefore, we present GD 12 results separately as concepti and no statistical comparison with placentas of later GDs was performed. The results are reported as Tukey boxplots (1.5-times IQR) of metabolite concentrations normalized to the placental protein content; *n* = 5 for each gestational day. Statistical significance was evaluated using the non-parametric Kruskal–Wallis test, followed by Dunn’s multiple comparisons test; * (*p* ≤ 0.05) and ** (*p* ≤ 0.01).

**Figure 2 ijms-21-07578-f002:**
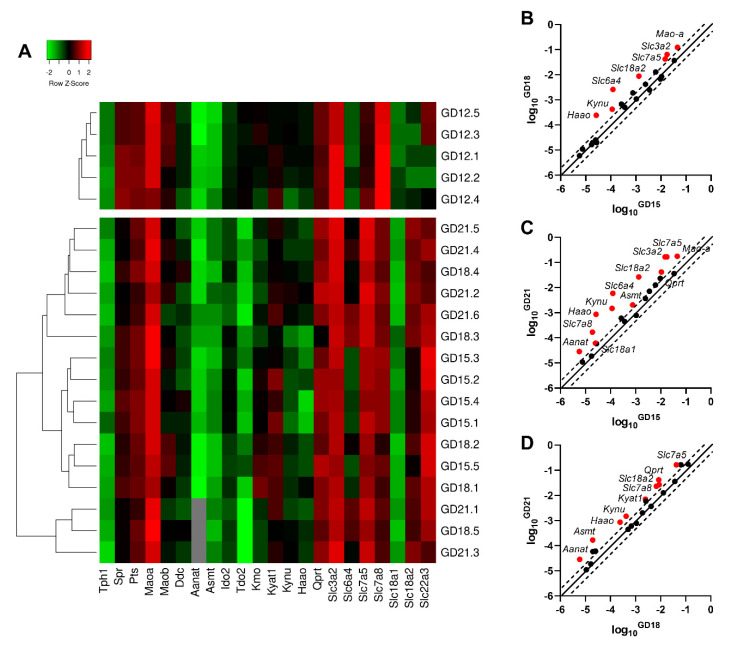
Gene expression of the main enzymes and transporters involved in TRP metabolic pathways in the rat placenta at different stages of gestation. (**A**) Heatmap representing qPCR gene expression analysis in rat concepti (GD 12) and the rat placenta (GD 15, 18, and 21). Average linkage clustering with Euclidean distance measurement reveals changes in gene expression that reflect the different stages of placental development. The color intensity indicates expression levels: red = upregulation; green = downregulation; and gray = not detected. (**B**–**D**) Identification of enzymes/transporters with significant changes in placental gene expression during gestation. Whilst the pattern of the increase in gene expression is similar for different stages of gestation, the highest difference is observed between GD 15 and 21. Data are presented as scatter plots with log10 gene expression at GD 18 compared to GD 15 (**B**), GD 21 compared to GD 15 (**C**), and GD 21 compared to GD 18 (**D**). The central diagonal line shows unchanged gene expression, and the dotted lines depict the threshold fold regulation (=2). Data were further evaluated using the non-parametric Mann–Whitney test on ΔCt values, and those which exceeded the threshold fold change (FC) and were statistically significant are highlighted in red and labeled.

**Figure 3 ijms-21-07578-f003:**
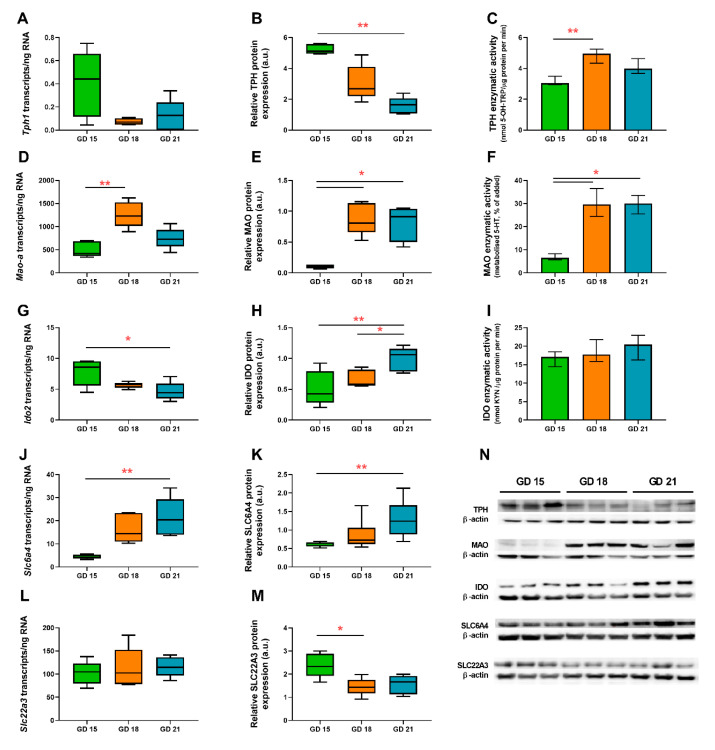
Expression and functional analysis of the rate-limiting enzymes: *Tph1*/tryptophan hydroxylase (TPH) (**A**–**C**), *Mao-a*/monoamine oxidase (MAO) (**D**–**F**), *Ido2*/indoleamine 2,3-dioxygenase (IDO) (**G**–**I**), and key transporters—*Slc6a4*/SLC6A4 (**J**,**K**) and *Slc22a3*/SLC22A3 (**L**,**M**)—of the TRP metabolic pathway in the rat placenta during gestation. Absolute quantification of the number of transcripts was evaluated by digital droplet PCR (**A**,**D**,**G**,**J**,**L**), whereas protein expression was evaluated by western blot analysis (**B**,**E**,**H**,**K**,**M**). Protein expression was normalized to β-actin as a loading control; representative immunoblots for target proteins and β-actin are shown (**N**). Enzymatic activity of TPH (**C**), MAO (**F**), and IDO (**I**) was evaluated as described in the Section 4. Data are presented as Tukey boxplots (1.5-times IQR) or the median with IQR; *n* = 5 for each gestational age. Statistical significance was evaluated using the non-parametric Kruskal–Wallis test, followed by Dunn’s multiple comparisons test; * (*p* ≤ 0.05) and ** (*p* ≤ 0.01).

**Figure 4 ijms-21-07578-f004:**
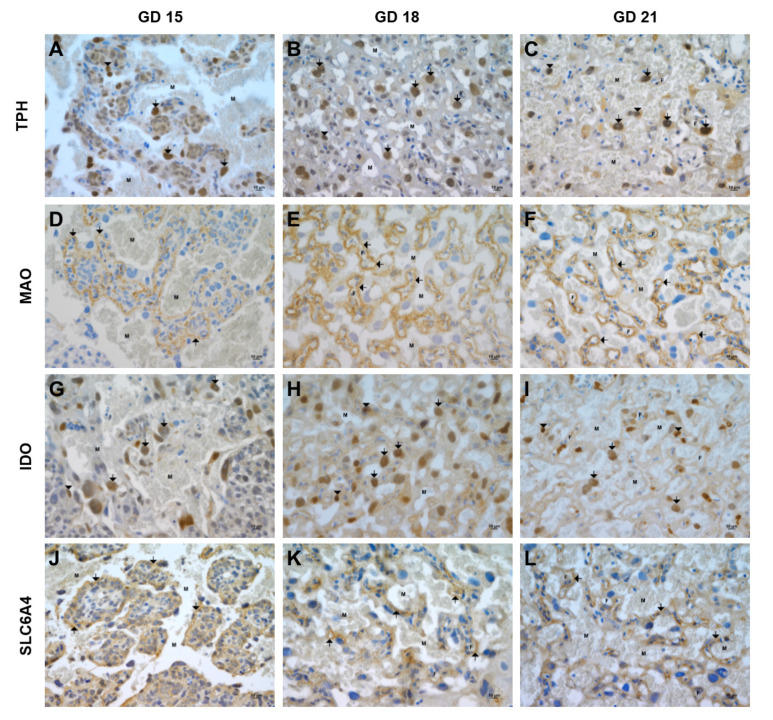
Immunohistochemical staining of TPH, MAO, IDO, and SLC6A4 in the rat placenta during gestation. TPH expression was detected in the labyrinth zone in cytotrophoblast cells (arrows) and fetal capillaries (arrowheads) at GD 15 (**A**), 18 (**B**), and 21 (**C**). MAO expression was only detected in the area of the labyrinth zone in cytotrophoblast/syncytiotrophoblast cells (arrows) at GD 15 (**D**) and 18 (**E**) and predominantly in the inner layers of the syncytiotrophoblast (layer II and III) at GD 21 (**F**). The most significant reactivity for IDO was visible in the labyrinth zone in cytotrophoblast cells (arrows) and fetal endothelial cells (arrowheads) at GD 15 (**G**), 18 (**H**), and 21 (**I**). Weak SLC6A4 expression was detected in cytotrophoblast/syncytiotrophoblast cells (arrows) in the placental labyrinth at GD 15 (**J**) and 18 (**K**), whereas at GD 21 (**L**), positivity was visible in the inner layers of the syncytiotrophoblast (layer II and III). M, maternal compartment, and F, fetal compartment. Bar: 10 μm.

**Figure 5 ijms-21-07578-f005:**
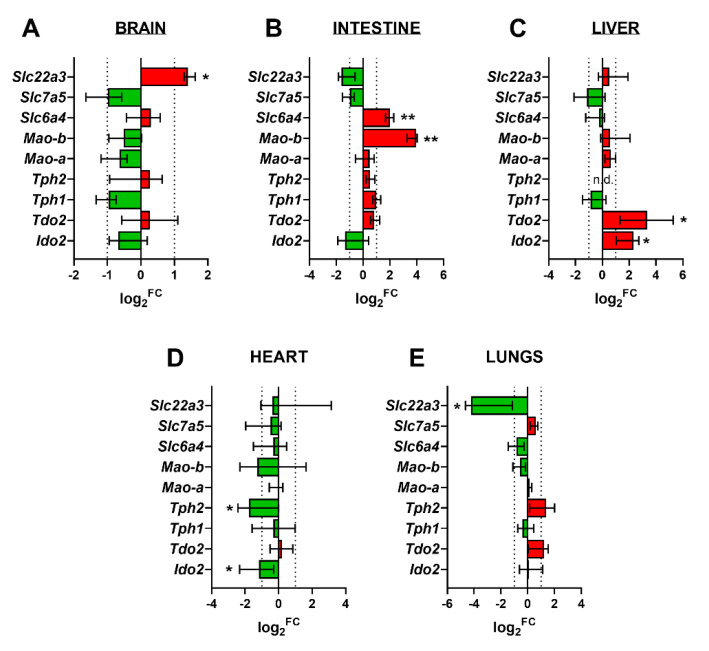
Gene expression of the main enzymes/transporters of the 5-HT and KYN pathways in rat fetal organs. Log2 fold change (FC) of gene expression at GD 21 compared to GD 18 is shown for the fetal brain (**A**), intestine (**B**), liver **(C**), heart (**D**), and lungs (**E**). Red color indicates upregulation, whereas green color indicates downregulation. Data are presented as the median with IQR; *n* = 5 for each gestational age. Statistical analysis of gestational age changes in the mRNA expression of target genes was evaluated using the non-parametric Mann–Whitney test; * (*p* ≤ 0.05) and ** (*p* ≤ 0.01). N.D., not detected.

**Figure 6 ijms-21-07578-f006:**
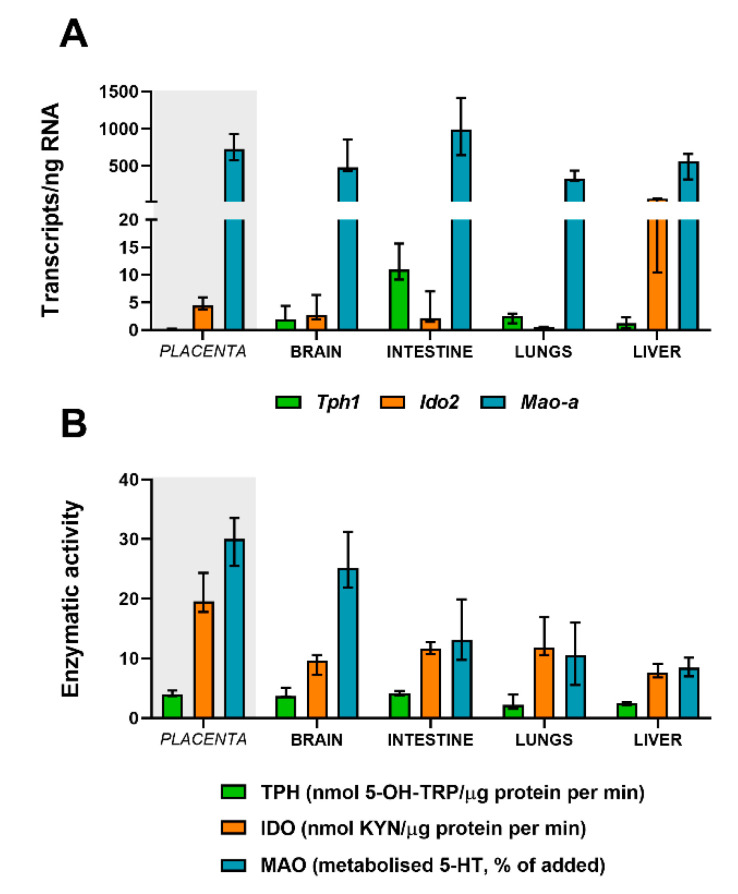
Gene expression and activity of *Tph1*/TPH, *Ido2*/IDO, and *Mao-a*/MAO in fetal organs. (**A**) Gene expression evaluated by digital droplet PCR analysis of the rate-limiting enzymes for the 5-HT pathway (*Tph1* and *Mao-a*) and KYN pathway (*Ido2*) at GD 21. (**B**) Functional analysis of TPH, IDO, and MAO enzymatic activity in fetal organs at term. Data are presented as the median with IQR; *n* = 5 for each gestational age.

**Figure 7 ijms-21-07578-f007:**
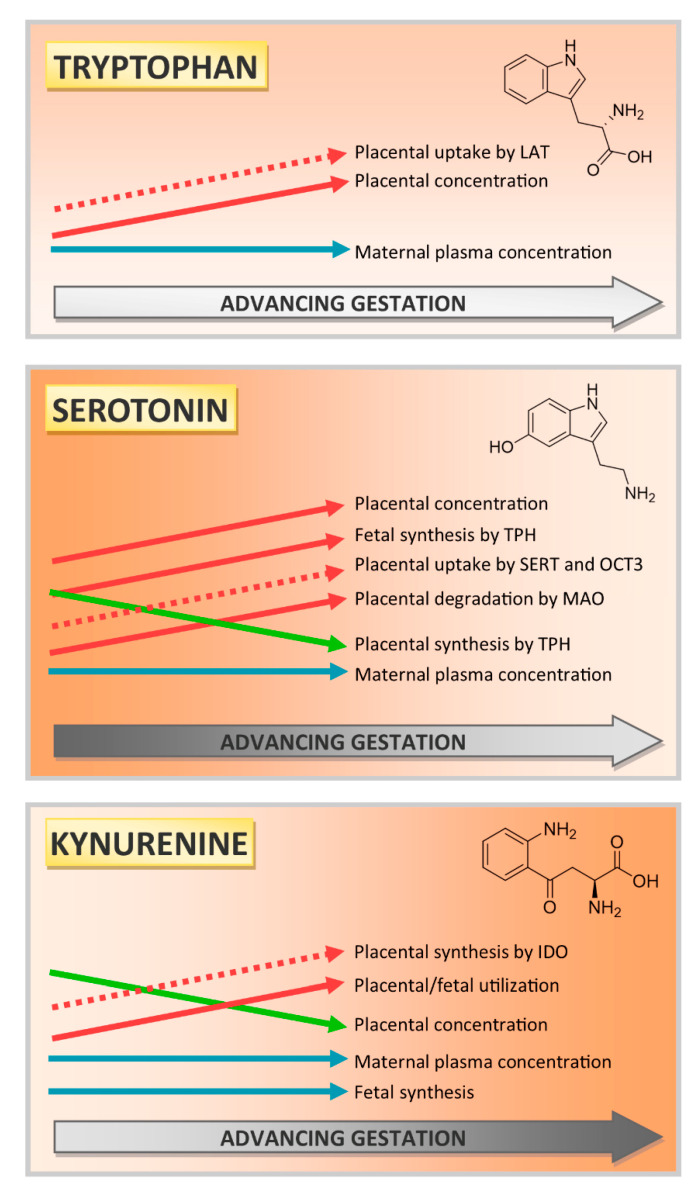
Schematic depiction of the TRP metabolic pathways in the rat fetoplacental unit during gestation. Although basal levels of TRP, 5-HT, and KYN in maternal plasma remain stable during gestation, their concentrations change significantly in the placenta and fetal organs. Based on our results and literature research [19,32,33,34,35,36,37] we show that the placental metabolism of TRP to 5-HT is important up to mid-gestation, with a subsequent decline (as the fetus becomes independent of placental sources of 5-HT). On the other hand, the placental production and utilization of KYN increase during pregnancy. Full arrows show metabolic activity or concentration levels, and dotted arrows show the gene or protein expression.

**Table 1 ijms-21-07578-t001:** Descriptive Statistics of the Pregnant Wistar Rats Used in the Study and Biochemically Assayed Metabolite Concentrations in Maternal Plasma. Results Are Expressed as the Mean ± SD; *n* = 5.

Parameter	GD 12	GD 15	GD 18	GD 21
Number of fetuses (*n*)	NA	11.33 ± 4.61	10.50 ± 2.89	8.25 ± 5.50
Rat weight (g)	333.00 ± 39.15	413.00 ± 54.27	358.00 ± 27.06	405.00 ± 5.00
Placental weight (g)	1.34 ± 0.25 ^1^	1.70 ± 0.36	3.58 ± 0.72	5.76 ± 1.96
Fetus weight (g)		1.75 ± 0.61	-	-
TRP plasma (µg/mL)	16.83 ± 2.75	19.58 ± 3.05	19.60 ± 1.64	13.16 ± 5.34
5-OH-TRP plasma (ng/mL)	78.81 ± 12.60	68.82 ± 7.94	76.04 ± 12.98	100.82 ± 37.76
5-HT plasma (ng/mL)	51.78 ± 41.10	154.98 ± 171.90	292.83 ± 314.57	215.59 ± 295.30
KYN plasma (ng/mL)	456.88 ± 213.10	407.82 ± 50.35	504.21 ± 98.76	429.93 ± 130.39
KYNA plasma (ng/mL)	155.60 ± 11.60	159.77 ± 53.80	147.20 ± 44.23	148.41 ± 26.38
5-OH-TRP/TRP ratio × 10^3^	4.73 ± 0.75	3.62 ± 0.86	3.94 ± 0.93	9.25 ± 5.61
KYN/TRP ratio × 10^3^	26.55 ± 9.69	21.03 ± 2.76	26.12 ± 6.91	36.33 ± 15.75
KYNA/KYN ratio × 10^3^	395.90 ± 151.48	410.26 ± 202.86	308.64 ± 138.30	359.73 ± 78.22

^1^ Concepti (consisting of embryos and extra-embryonic tissues).

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
