# Peer review of "Profiling of Tryptophan Metabolic Pathways in the Rat Fetoplacental Unit during Gestation"

_ijms, 2020, doi:10.3390/ijms21207578_

Round 1
Reviewer 1 Report
The article is devoted to the study of placental homeostasis of tryptophan, which plays a decisive role in the development and programming of the fetus. The authors performed a comprehensive study of the metabolic pathways of tryptophan using an animal model – Wistar rats. The authors presented a dynamic study of the metabolites concentration at different stages of gestation. A wide range of biochemical analytical methods were used in the work. The authors discovered an interesting fact that Ido1 is not expressed in the placenta, as well as in fetal tissues.
The conclusion of this work is that any insult, including pathological conditions, environmental factors, medication, and/or epigenetics during pregnancy may significantly affect its orchestration resulting in suboptimal in utero conditions.
The obtained results are rather interesting and important both for fundamental biology and for application in biomedicine.
The manuscript is written in clear, concise style, and is easy to follow. The experimental design is well thought out, and the statistical analysis is performed at the modern level. I have just a few critical comments:
- Comment to Table 1. The data values presented incorrectly. Please provide values according to the significant figure rules. What does the factor "103" mean in the last three lines? Apparently, this is a raised number?
- Lines 109-110 «In a silimar fashion»: The plots shown in Figures1 C and F are not similar. Figure1 F shows growth of concentration from GD12 to GD21, while in Figure1 C the character of the data changes is another. The concentration maximum is observed on GD18, and then the concentration drops noticeably. Please explain this situation.
Author Response
Thank you very much forhaving read our manuscript carefully and for yourthoughtfulcomments.
1) Comment to Table 1. The data values presented incorrectly. Please provide values according to the significant figure rules. What does the factor "103" mean in the last three lines? Apparently, this is a raised number?
Answer: We apologize for this mistake;it occurred when we adjusted our manuscript to IJMS template. When calculating ratios of metabolite to precursor, the values are numerically small, thus, we multiplied them by the factor 103, which was converted to plain text as “103”.We have corrected this errorin the revised version of our manuscript.
2) Lines 109-110 «In a similarfashion»: The plots shown in Figures1 C and F are not similar. Figure1 F shows growth of concentration from GD12 to GD21, while in Figure1 C the character of the data changes is another. The concentration maximum is observed on GD18, and then the concentration drops noticeably. Please explain this situation.
Answer: We agree with the comment-we have amended the manuscript correspondingly; see line 105 in the revised manuscript.
Reviewer 2 Report
The study of metabolites that are involved in the development of the placenta and can affect the fetus is very important and promising in reproductive biology. An adequate animal model allows to find the main metabolites that can affect fetal development. In this study, the authors evaluate two main metabolic pathways of tryptophan in pregnant rats. They use modern methods to evaluate metabolites and understand interaction pathways. However, the authors do not indicate aim of their research and, therefore, cannot draw the main conclusions. This is a significant drawback of this work. The reader does not understand what conclusion the authors of this article came to and how this result can be applied to humans and to reproductive biology in general. It is not clear what happens to the fetus when the metabolites of tryptophan change.
Major comments and Suggestions for Authors
Line 23 The term “tryptophan handling” is not clear, please replace to metabolism or similar.
Line 34 The phrase “the developmental origins of health and disease” is not suitable. Rewrite please.
Line 82-93 One or two sentences should be included to explain the main purpose of this experiment. The authors indicate only the purpose of using experimental model and main steps of the work, but have not indicate aim of this work.
Line 96 It is indicated that “concentrations in rat placenta and maternal plasma” but in fact the authors wrote only about plasma. Improve or include data about placenta.
Line 96-101 The authors should include data on the number of developed and undeveloped embryos in this paragraph or in table 1.
In table 1 The last row of data “KYN/KYNA ratio x 103” include a mistake, because the mean KYN divided to the mean KYNA cannot be 395.9 This will be about 3-4. The same errors are in next columns.
In table 1 in the row “5-HT plasma” in the column GD15, GD18 and GD21 SD is higher than means. How can you comment these facts?
Could you show the same metabolites in non-pregnant dams? It could be well reference for understanding changes these metabolites in pregnant.
Did you compare GD15, GD18 and GD21 with GD12 on the figure 1? There was shown strong effect on the same metabolites when comparing GD12 and GD18, but no statistical data are available.
As for the statistical assay: why did you use KW test (as described in M&M) but this statistical test was not included in the text.
At the end of each chapter in "Results" need to add the short conclusion (1-2 sentence).
In the figure 2 in GD21 n=6, but in text (line 120) the authors wrote that n=5.
Line 166 The Authors suggest that “preferential TRP metabolism toward the KYN pathway at the final stages of rat pregnancy”, but in the figure1 KYN has been reduced to GD21. How did the authors make this suggestion?
In the figure 3A expression of Tph1 on the GD18 was lower but no significant statistical test is presented. Is this a mistake or really a minor effect?
Comparison statistics data presented on immunohistochemical sections of the placenta should be included. Data need to present as mean+-SD.
Please include main results at the end of the discussion. Difficult to understand main idea of this article.
Line 379-391 there are not information about laboratory animal health status, the rat ages, how to housekeep of rat dams (with or without males?)
Please wrote to M&M which methods were used to assay each metabolite.
Line 515-519 Please wrote which statistical test was used for each assay data. Why don’t authors use the KW test to compare groups GD15, GD18 and GD21?
Author Response
Thank you indeed for having read our manuscript carefully and for pointing out several issues,
which are addressed below:
1) Line 23 The term “tryptophan handling” is not clear, please replace to metabolism or
similar.
Answer: Thank you for your comment. Based on your suggestion, we have replaced it with
“tryptophan pathways” to include both metabolism and transport systems (line 23).
2) Line 34 The phrase “the developmental origins of health and disease” is not suitable.
Rewrite please.
Answer: We agree with your suggestion; we have changed it to “…have a negative impact on
fetal programming” (line 34).
3) Line 82-93 One or two sentences should be included to explain the main purpose of this
experiment. The authors indicate only the purpose of using experimental model and main
steps of the work but have not indicate aim of this work.
Answer: Based on your suggestion, we have amended the final paragraph of the Introduction
(lines 82 - 89). Now it reads: “We have recently reported that human placental catabolism of
TRP is subject to developmental changes during pregnancy and trophoblast differentiation
[33]. In this study, we hypothesized, that apart from the placenta, fetal organs also contribute
to the overall TRP homeostasis in the fetoplacental unit. Since experiments in pregnant women
are limited due to ethical and technical reasons, current research depends on the use of
alternative approaches, including experimental animals. Therefore, here we used Wistar rat as
the most suitable model for placental TRP metabolism in health and disease [3] to provide indepth evidence on regulation of the TRP homeostasis in the fetoplacental unit during
gestation.”
4) Line 96 It is indicated that “concentrations in rat placenta and maternal plasma” but in fact
the authors wrote only about plasma. Improve or include data about placenta.
Answer: The results on placental concentrations of metabolites are included below Table 1.
Specifically, the descriptive results are presented in lines 101 – 109 and also in Figure 1.
5) Line 96-101 The authors should include data on the number of developed and undeveloped
embryos in this paragraph or in table 1.
Answer: In Table 1 we presented the mean number of fetuses for each gestation age. In this
study we worked only with pregnancies having all embryos developed.
6) In table 1 The last row of data “KYN/KYNA ratio x 103” include a mistake, because the
mean KYN divided to the mean KYNA cannot be 395.9 This will be about 3-4. The same
errors are in next columns.
Answer: Thank you for spotting this inconsistency. The KYN/KYNA ratio was corrected into
the right form (KYNA/KYN). Furthermore, the mistake of “x 103” occurred when we adjusted
our manuscript to IJMS template. When calculating ratios of metabolite to precursor, the values
are numerically small, thus, we have multiplied them by the factor 103
, which was converted to
plain text as “103”. We have corrected this error in the revised version of our manuscript.
7) In table 1 in the row “5-HT plasma” in the column GD15, GD18 and GD21 SD is higher
than means. How can you comment these facts?
Answer: Indeed, plasma 5-HT concentrations showed high standard deviations from the mean.
The only plausible explanation we have is that there is high interindividual variability
accounting for the differences, which we refer to in lines 96 - 97.
8) Could you show the same metabolites in non-pregnant dams? It could be well reference for
understanding changes these metabolites in pregnant.
Answer: The differences in TRP concentrations between pregnant and non-pregnant women
have been described in the literature [1, 2]. The aim of our study was to describe and quantify
the physiological, gestational-age dependent changes of tryptophan metabolism and transport,
focusing on placenta and fetal organs; determination of metabolite levels in maternal plasma
during gestation served to asses possible systemic changes independent of the fetoplacental
unit.
In non-pregnant dams, the only unit available for measurement is maternal plasma, which may
be a useful indicator when comparing physiological and pathological (e.g. disease) states.
Nonetheless, for this study it would require additional animals to be sacrificed and we believe
that this information would be of no or limited added value to our manuscript.
9) Did you compare GD15, GD18 and GD21 with GD12 on the figure 1? There was shown
strong effect on the same metabolites when comparing GD12 and GD18, but no statistical
data are available.
Answer: Statistical analysis against the concepti samples (GD 12) was not performed as it
represents a mixed tissue (embryonic and extra-embryonic) and comparing it with placenta
(extra-embryonic tissue) could provide wrong information. Nonetheless, based on your
comment we have added this information in the legend of Figure 1 (lines 110 – 113); it now
reads “It is important to note that at GD 12, the rat placenta is technically indistinguishable
from the embryo; therefore, we present GD 12 results separately as concepti and no statistical
comparison with placentas of later GDs was performed.”
We have also improved Figure 1 so that the lines separating GD 12 (concepti sample consisting
of embryonic and extra-embryonic tissue) and GD 15, 18 and 21 (placenta samples) are clearly
visible.
10) As for the statistical assay: why did you use KW test (as described in M&M) but this
statistical test was not included in the text.
Answer: Since the data were not always normally distributed, we used the non-parametric
Kruskal-Wallis test to compare between more than two groups. The information on statistical
method used is clearly stated in the Materials and Methods section (lines 554 - 556), as well as
in each figure legend.
11) At the end of each chapter in "Results" need to add the short conclusion (1-2 sentence).
Answer: Where appropriate, we conclude the Result paragraphs with a short conclusion; we
have read several other papers published in IJMS to adhere to the house style. See e.g. lines
156 - 158, 165 - 167.
12) In the figure 2 in GD21 n=6, but in text (line 120) the authors wrote that n=5.
Answer: Thank you for your comment; it was a mistake on our side. The qPCR data included
6 samples for GD 21 and this information was corrected in the revised version of our manuscript
(lines 118 - 120).
13) Line 166 The Authors suggest that “preferential TRP metabolism toward the KYN pathway
at the final stages of rat pregnancy”, but in the figure1 KYN has been reduced to GD21.
How did the authors make this suggestion?
Answer: This is an interesting point; however, this short conclusion is referring to the protein
expression of IDO, the rate-limiting enzyme for KYN pathway. IDO protein expression showed
an increased trend toward GD 21, compared to GD 15 and 18 (Figure 3H). Nevertheless, it is
true that the placental content of KYN decreased significantly towards term. Interpretation of
these findings is provided in the Discussion section (lines 372 - 377) where we hypothesize two
possible scenarios:
• The decrease in KYN content might be attributed to the rapid utilization of KYN for
synthesis of other metabolites inside the placenta. Indeed, we found that gene expression of
enzymes downstream KYN pathways show upregulation in GD 21 compared to GD 15.
Additionally, we detected a steady increase in placental KYNA content from GD 15 to 21.
• KYN can be further transported to fetal circulation for synthesis of KYNA and other
neuroactive and immunomodulatory metabolites in the fetal organs.
14) In the figure 3A expression of Tph1 on the GD18 was lower but no significant statistical
test is presented. Is this a mistake or really a minor effect?
Answer: Due to the high interindividual variability in GD 15, this effect is indeed minor and
statistically insignificant.
15) Comparison statistics data presented on immunohistochemical sections of the placenta
should be included. Data need to present as mean+-SD.
Answer: For the detection of our proteins of interest, we designed a redundant experimental
strategy taking advantage of the strengths of immunoblotting (western blot) [3] and
immunohistochemistry [4] techniques. Even though both techniques could be used for protein
quantification, the immunoblot has a higher sensitivity for protein quantification compared to
the immunohistochemistry technique [5].
The main aim of the immunohistochemistry analysis was to describe the co-localization of the
key enzymes and transporters involved in TRP metabolism (TPH, MAO, IDO and SLC6A4) in
rat placenta across pregnancy; the use of this technique is justified since it allowed for native
localization of the proteins of interest in 2D placental tissues.
For quantifications of the relative protein expression (TPH, MAO, IDO, SLC6A4 and
SLC22A3) we performed Western blot analysis and results are shown in Figure 3 (B, E, H, K
and M), presented as Tukey boxplots with appropriate statistical evaluation applied. We believe
that this successful overlapping strategy allowed us to present robust results.
16) Please include main results at the end of the discussion. Difficult to understand main idea
of this article.
Answer: Thank you for your helpful suggestion. In the revised manuscript, we have connected
Discussion and Conclusions into one section so that the flow of the text is easier to follow for
readers (see lines 401 - 409); it now reads: “In conclusion, our results in Wistar rat demonstrate
that under physiological conditions placental TRP metabolism to 5-HT is crucial up to midgestation. Afterwards, the fetus is capable of 5-HT synthesis and becomes independent of
maternal and/or placental sources. Moreover, at term, the placenta and fetal organs
orchestrate 5-HT homeostasis via the activity of transporters and metabolizing enzymes, likely
to prevent hyper/hypo-serotonemia in the fetoplacental unit. On the other hand, placental
production of kynurenine increases during pregnancy with a low contribution of fetal organs
throughout gestation (Figure 7). Importantly, our data obtained from the rat placenta are in a
close agreement with those observed in humans [33], confirming Wistar rat as an appropriate
model for further studies on TRP homeostasis in the fetoplacental unit.”
In addition, we have included a summarizing Figure 7, to make the main outcomes of our study
visible and understandable for all readers.
17) Line 379-391 there are not information about laboratory animal health status, the rat ages,
how to housekeep of rat dams (with or without males?)
Answer: All important information on animal housing are provided in Materials and Methods;
following your suggestion, we have added the age of the rats. See lines 417 - 424 of the revised
manuscript.
18) Please wrote to M&M which methods were used to assay each metabolite.
Answer: All metabolites were assayed by HPLC, described in detail (including excitation and
emission wavelengths of fluorescence detector and wavelengths of UV detector for each
metabolite) in lines 530 - 539.
19) Line 515-519 Please wrote which statistical test was used for each assay data. Why don’t
authors use the KW test to compare groups GD15, GD18 and GD21?
Answer: Thank you for your remark – indeed the Kruskal-Wallis test is more appropriate for
comparing groups and in the revised version of the manuscript the analysis for Figure 3 is
changed to Kruskal-Wallis test. The Mann-Whitney tests are used only when we compared
between two groups (like in Figure 2, 5 and 6).
To address this issue we have added an explanation in lines 554 - 558, which now reads:
“Experimental outcomes were assessed using non-parametric Mann-Whitney tests (when
comparing two groups) or Kruskal-Wallis followed by Dunn’s multiple comparisons test (when
comparing more than two groups).” Moreover, each Figure legend includes the information on
the statistical test performed.